# Females of *Halyomorpha halys* (Hemiptera: Pentatomidae) Experience a Facultative Reproductive Diapause in Northern Greece

**DOI:** 10.3390/insects13100866

**Published:** 2022-09-23

**Authors:** Eleni I. Koutsogeorgiou, Nikos A. Kouloussis, Dimitrios S. Koveos, Stefanos S. Andreadis

**Affiliations:** 1Laboratory of Applied Zoology and Parasitology, School of Agriculture, Aristotle University of Thessaloniki, 54124 Thessaloniki, Greece; 2Institute of Plant Breeding and Genetic Resources, Hellenic Agricultural Organization “DEMETER”, 57001 Thermi, Greece

**Keywords:** reproductive diapause, facultative, preoviposition period, fecundity, egg masses, brown marmorated stink bug

## Abstract

**Simple Summary:**

In this study we investigated the oviposition rates and fecundity of females of *Halyomorpha halys*, by collecting adults soon after they entered their overwintering sites and allowing them to overwinter for 2, 3, 4 and 5 months in natural conditions of Northern Greece. We studied the occurrence of a facultative reproductive diapause in females of this insect in N. Greece by measuring the preoviposition period. According to our results, females of *H. halys* need an additional time period to exit diapause after they emerge from overwintering sites to mature sexually and start laying eggs. This finding is especially important to assess the possible damage of *H. halys* to early maturing crops for any forecasting models and thus management strategies.

**Abstract:**

*Halyomorpha halys* (Stål) (Hemiptera: Pentatomidae) is a native pest of East Asia that overwinters as an adult in natural and human-made structures. Adult emergence from overwintering sites starts in spring, whereas females produce offspring in early summer on host plants, where most feeding occurs. In this study, we investigated the reproductive physiology of overwintering females of *H. halys* in Northern Greece, by determining the duration of the preoviposition period and fecundity of individuals that were left to overwinter in natural conditions and were subsequently transferred to chambers with standard conditions monthly, from December 2020 to March 2021. According to our results, overwintering *H. halys* females do not initiate egg laying once they emerge from overwintering sites, but rather need some additional time to exit diapause and mature reproductively. The mean preoviposition period of overwintering females that were transferred from their overwintering sites to the chambers in December 2020 was 29.0 days, which was significantly longer by 8.3 days than that of females that overwintered until March 2021, and by 13.2 days than the control (26 °C, 60% RH and a 16:8 h light: dark photoperiod). No significant difference among the average number of eggs per egg mass laid by overwintering individuals brought in the chambers in different time intervals and the laboratory colony was observed. However, females that were left to overwinter until March laid a significantly higher number of eggs in total, compared to the ones whose overwintering was disrupted in February. Based on our findings, overwintering females of *H. halys* experience a facultative reproductive diapause in Northern Greece. Our study was the first to determine the occurrence of diapause of *H. halys* in N. Greece and our findings could be very valuable for assessing the damage of this pest to early-season crops and designing successful management practices.

## 1. Introduction

The brown marmorated stink bug (BMSB), *Halyomorpha halys* Stål (Hemiptera: Pentatomidae) is a native pest of eastern Asia that was accidentally introduced into the United States in the mid-1990s. It was first detected in Pennsylvania [1] and since then has spread throughout much of the United States and Canada [2]. In Europe, it was first reported in Liechtenstein in 2004 [3] and in Zurich, Switzerland in 2007 [4]. In the US, *H. halys* is considered a generalist herbivore, with a range of more than 100 different host plants, often resulting in substantial economic damage [5]. In Greece, the first occurrence of *H. halys* was reported in 2011, as a nuisance pest in houses in the center of Athens [6], while in 2017 the first serious damages by *H. halys* were recorded in kiwi orchards (*Actinidia chinensis*) [7]. However, despite it being an extremely widespread pest, little is known about its biology and its phenology in Greece. Due to its ability to invade and colonize new habitats, its high dispersal capacity and polyphagy, it poses a major threat to Greece’s agriculture, especially to major crops such as orchard fruits (kiwi, apple, peach), as well as vegetable crops (tomato, bean) which are a great part of the primary agricultural production of Greece.

In Asia, *H. halys* has been recorded to complete 4–6 generations per year [8]. It overwinters as an adult and produces offspring in the midsummer, when most of the feeding damage occurs [8]. In the mid-Atlantic US states and Russia, it is considered bivoltine [9,10], and adults overwinter in concealed, sheltered locations, underneath bark of trees or human-made structures [11], within which thousands of individuals cluster together [12]. In these regions, *H. halys* adults begin to emerge from overwintering sites in April [12]. However, their feeding behavior immediately after emerging from overwintering sites is still unknown, since host resources are limited, although in Asia, they are considered to attack tree hosts [13,14]. The distribution of the overwintering population among these sheltered sites is also unknown, and the survivorship of those individuals that overwinter in natural settings is likely influenced by annual variations in winter temperatures, the thermal buffering capacity of specific sites and the potential microhabitat effects created by their aggregations [12]. *Halyomorpha halys* was characterized as cold-intolerant by Cira et al. [15] in a study examining the supercooling point of the overwintering population, as mortality rapidly increased at temperatures below −10 °C. Laboratory studies have shown a preference of *H. halys* to a dark refuge rather than a lighted one, indicating that the selection of overwintering sites is affected by photosensitivity [16].

Adults of *H. halys* overwinter in a non-feeding, non-reproductive state [9]. Overwintering females of *H. halys* are considered previtellogenic [17] and require an additional period of development prior to sexual maturity. In Japan, the critical photoperiod for ovarian development is between 13.5 and 14.0 h [18], while in temperate climates, overwintering *H. halys* adults remain in a physiological state of diapause before initiation of reproduction [19]. Diapause is a dynamic state of low metabolic activity mediated by the neuro-hormonal system of insects [20]. Depending on the developmental stage, different hormones and molecular signaling pathways regulate entry and exit from diapause. Inability of the corpora allata to produce the juvenile hormone required for reproduction is commonly what determines adult diapause. Insulin signaling is a key component of diapause, and transcription factors offer a mechanism to explain how a single hormone response may result in multiple downstream consequences triggered by diapause [21]. In a study evaluating survival and fecundity of *H. halys* adults, Taylor et al. [22] stored field collected, presumably dormant individuals at 9 °C in complete darkness, before maintaining them at 25 °C and recorded that females did not initiate egg laying within the time period that nondormant individuals would be expected to start laying eggs at 25 °C and 16:8 h L:D, as stated by Nielsen et al. [9] and Haye et al. [23]. Nonetheless, Taylor et al. [22] did not state the photoperiod that individuals were maintained at 25 °C, thus it remains unclear whether this delay in oviposition was induced by the photoperiod in this experiment or by morphogenetic changes from diapause [19].

The response of *H. halys* to low temperature exposure could affect its ability to acclimate to temperate climates and alter its behavior post-emergence from overwintering sites. This study aims to investigate the reproductive physiology of overwintering females of *H. halys* in Northern Greece by determining their preoviposition period and evaluating their fecundity. The response of *H. halys* populations to these conditions could provide information and guidance to assess the risk of seasonal crop damage after emergence from overwintering sites.

## 2. Materials and Methods

### 2.1. Specimen Collection

Approximately 3000 overwintering males and females of *H. halys* were hand collected in sleeve cages of polyethylene net (35 × 15 cm, 60 mm mesh grid) (Suxess, IKE, Greece) from sheltered overwintering sites on the 6th and 7th of October 2020 in Naoussa, Greece (40°37′21.12″ N, 22°2′23.04″ E). Adults were maintained in groups of 100 in 90 × 60 × 60 cm mesh cages (Raising Butterflies, Salt Lake City, UT, USA) in natural conditions at the Institute of Plant Breeding and Genetic Resources in Thermi, Thessaloniki, Greece (40°32′18″ N, 22°59′57″ E), protected from direct sunlight or heavy rain, but exposed to outdoor conditions (photoperiod, temperature and relative humidity), during October 2020 to March 2021. Cardboard paper was inserted inside the cages to simulate overwintering sites. Temperature within overwintering units was monitored using data loggers (HOBO U23 Pro v2 Temperature/Relative Humidity Data Logger, Onset Computing, Bourne, MA, USA).

### 2.2. Colony Maintenance

The laboratory colony was maintained as described by Andreadis et al. [24]. Adults and nymphs were collected from abandoned buildings and fields in central Macedonia. Insects were transferred to separate mesh cages (30 × 30 × 30 cm) with vinyl window and zip closure (Raising Butterflies, Salt Lake City, UT, USA) based on their developmental stage. Both nymphs and adults were provided with water and green beans (*Phaseolus vulgaris*) and maintained in the laboratory at 26 °C, 60% RH and a 16:8 h light:dark photoperiod. In addition, adults were provided with green bean seedlings for oviposition. Adult females typically laid eggs on the underside of the bean leaves, or on the top or side of the mesh cage. Egg masses were removed every other day and placed on the top of a small green bean leaf which was attached to a 4 mL clear screw vial (45 × 14.7 mm, BGB, Germany) that was glued to the bottom of a 460 mL cylindrical clear plastic cup (9 cm diameter × 7 cm height). Cotton balls soaked in water were added into the cups to increase humidity.

After egg hatching, first instar nymphs remained aggregated in clusters at the top of the green bean leaf until molting into second instar nymphs. Second instar nymphs were placed in new cylindrical clear plastic cups with the use of a size 0 artist paint brush (Sapphire, Michaels, Irving, TX, USA), along with a single green bean pod and moistened cotton balls, where they remained until reaching the fifth instar. Feed was replaced twice a week. Afterwards, they were transferred into mesh cages to complete their development to adult. Fifth instar nymphs were transferred to the mesh cages using featherweight forceps (BioQuip Products, Rancho Dominguez, CA, USA) to avoid injury or death.

### 2.3. Bioassay

Starting from 30 December 2020, groups of 10 males and 10 females of the collected overwintering adults that were maintained in natural outdoor conditions were randomly selected and brought in standardized chambers (9 m^2^) with stable conditions (26 °C, 60% RH, 16:8 h light:dark photoperiod) monthly, until 30 March 2021. A total of 6 replicates were conducted for each month (120 individuals used in total: 60 females and 60 males). Adults were maintained in the chambers in 30 × 30 × 30 cm mesh cages (Raising Butterflies, Salt Lake City, UT, USA), fed on green beans (*Phaseolus vulgaris*) and a cotton ball soaked in 10% sugar water. Food was replaced every 2–3 days. A potted bean plant was used as an oviposition substrate. For each replicate, mortality and oviposition were monitored daily to determine the preoviposition period and fecundity of females. Individuals from an established laboratory colony (3 replicates of 10 males and 10 females in each) were used as control. The preoviposition period was calculated in days, as the time between the date that individuals were brought into the chambers until the date that the first egg mass was laid. The egg masses were removed daily from each cage and placed on a moist filter paper inside cylindrical plastic vials (9 cm diameter × 7 cm height), labeled by date to monitor successful hatching. Each replicate was monitored until the death of all males and all females.

### 2.4. Female Reproductive Status

To determine the reproductive status of the collected overwintering females, 10 individuals were dissected prior their transfer to the chambers each month, and their reproductive state was characterized according to the revised Nielsen ranking system [25]. Insect legs and wings were removed prior to dissection and the insect was placed ventral side- down on a Petri dish under a stereo microscope (SZ51, Olympus). Ethyl alcohol (C_2_H_5_OH) was used for all dissections to prevent desiccation. Overwintering females were characterized as previtellogenic (rank 2) and unmated (Figure 1).

### 2.5. Statistical Analysis

We used generalized linear models (GLM) with quasi-Poisson-distributed errors and long-link function to test the potential effect of interruption of the overwintering state on the preoviposition period, number of eggs per egg mass and total egg masses laid by *H. halys* females. Previous analyses revealed overdispersion when using the GLM models with Poisson-distributed errors. Means among the different preoviposition periods, number of eggs per egg mass and the total egg masses were compared using Tuckey’s test. All statistical tests incorporate a type-I error rate of a = 0.05, and all parametric statistics were carried out using R v. 3. 5.1 [26].

## 3. Results

*Halyomorpha halys* adults used in this study were subjected to the following weather conditions: 18.1 ± 0.5 °C (mean ± SEM) during the warmest month (October) and 8.2 ± 0.8 °C (mean ± SEM) during the coldest month (January) (Figure 2).

### 3.1. Preoviposition Period

We observed a statistically significant difference among the preoviposition period (in days, since transfer of females in chambers with stable conditions until the first egg mass laid) of individuals that were induced to emerge from overwintering in December 2020 and the ones that overwintered until March 2021, and the laboratory colony (control) (Figure 3) (quasi-Poisson GLM: *p* = 1.66 × 10^−6^). The preoviposition period of females that overwintered under natural outdoor conditions until December was 29.0 ± 2.0 days, whereas those who were left to overwinter under natural outdoor conditions until March initiated their oviposition in 20.7 ± 1.8 days. Females of the laboratory colony laid the first egg mass in 15.8 ± 0.3 days. There was no statistically significant difference among the preoviposition period of individuals that overwintered until January and February (24.5 ± 1.4 and 25.5 ± 2.1 days, respectively) (Figure 3). Females of the laboratory colony displayed the shortest preoviposition time compared to all other treatments, and this difference was found to be significant (Figure 3).

### 3.2. Fecundity of H. halys Females

There was no significant difference among the average number of eggs per egg mass laid by individuals that overwintered until December, January, February and March, and the laboratory colony (control) (Figure 4) (quasi-Poisson GLM: *p* = 0.165). Individuals that were left to overwinter in natural conditions until March laid a significantly higher number of eggs in total, compared to the ones whose overwintering was disrupted in February (quasi-Poisson GLM: *p* = 1.18 × 10^−4^) (Figure 5). Individuals of the laboratory colony (control) laid a significantly higher number of eggs in total compared to all other treatments (Figure 5).

## 4. Discussion

Our results indicated a statistically significant difference between the preoviposition periods of the laboratory colony (control) and all individuals that overwintered until December, January, February, and March and were then transferred to chambers with stable conditions. We also observed significantly different preoviposition periods between individuals that overwintered until December and March. These are in agreement with previous studies on the overwintering and reproduction of *H. halys* females; Nielsen et al. [9] and Rice et al. [14] both stated that females enter overwintering sites reproductively immature and unmated, they emerge from these sites in April and require an additional period of time for diapause termination before they initiate their oviposition, which is likely driven by photoperiod. However, diapause in *H. halys* is facultative and its induction and termination are affected by temperature and photoperiod, and that is what gives the ability to *H. halys* to easily adapt to regions with shorter maximum day lengths [25]. Facultative diapause is mediated by external factors and does not necessarily occur in each generation [27]. At Northern latitudes, above 30°, these cues trigger physiological changes and acclimation to winter temperatures [28,29]. Individuals whose overwintering was disrupted in December started laying eggs on average in 29 ± 2.0 days, whereas those who were left to overwinter in natural outdoor conditions until March initiated their oviposition in a significantly shorter time. This gradual decrease in the preoviposition time of adult females indicates that this species experiences facultative reproductive diapause in N. Greece. Our results are the first to provide knowledge on the reproductive physiology of this insect in this location and agree with a previous study by Haye et al. [23], where adults of *H. halys* that were field collected from their overwintering sites and stored in natural conditions during winter became active when daily maximum temperatures reached 25°C and above. When moved to an incubator at 25 °C and 16:8h light dark photoperiod, it took on average 20.1 ± 1.47 days until females started laying eggs. This confirms that females require some additional time to terminate diapause once they leave their overwintering sites. In recent works, Nielsen et al. [25] and Reznik et al. [10] also concluded that *H. halys* enters reproductive diapause in temperate locations in the fall and that a delay occurs in developmental maturity after diapause termination in the spring.

Fecundity and reproduction in insects can be influenced by abiotic factors such as light intensity and daylength [30] as well as biotic factors such as food [31,32]. Temperature is known to influence several biological characteristics in insects, including fecundity, fertility and photoperiodic response [27,33,34]. The preoviposition period of *Podisus nigrispinus* (Hemiptera: Pentatomidae) was 27.6 ± 0.9 days at 17 °C and gradually decreased with increasing temperatures, with the shortest being 7.5 ± 0.3 days at 29 °C [35]. Furthermore, the stink bug *Cyptocephala alvarengai* (Hemiptera: Pentatomidae) showed a preoviposition period ranging from 4 to 15 days with an average of 7 ± 1.1 days at 25 °C [36]. Regarding *H. halys*, Baek et al. [37] found that the preoviposition period also decreased with increasing temperatures, with that at 25 °C being 21.5 ± 2.2 days. This result agrees with our findings as well, although Nielsen et al. [9] showed a preoviposition period of 13.4 ± 0.7 days on average. However, as mentioned before, oviposition in insects can be furtherly affected by several abiotic and biotic factors except for temperature, including diet. Actually, the preoviposition period of adult females of *Dichelops furcatus* (Hemiptera: Pentatomidae) was 11.6 ± 0.4 days on soybean, 24.1 ± 1.7 days on oat, and 20 ± 6.0 days on rye [38].

The average number of eggs laid per egg mass in our study ranged from 25.2 to 27 eggs per egg mass, with no statistically significant differences among treatments and concurs with a previous study by Dingha and Jackai [39], who established a laboratory colony of *H. halys*, maintained it at 27 ± 2 °C, 70 ± 10% RH, 16:8h light dark photoperiod and on a combination of food substrates and counted 26.0 ± 2.1 eggs per egg mass. On the other hand, we observed significant differences between the total number of eggs laid by individuals that emerged from overwintering sites in December, January, and February, and those who were left to overwinter until March. Females of the laboratory colony laid significantly more eggs in total compared to all other treatments. Likewise, adults of *H. halys* displayed highest fecundity at constant temperatures exceeding 17 °C [40]. Our results indicate that while the collected overwintering females were fecund and capable of producing offspring, their oviposition was possibly influenced by abiotic factors such as temperature and photoperiod, as also suggested by Musolin et al. [41]. It is possible that this exposure to low temperatures is needed to produce a high number of eggs in total. Long-term exposure of *H. halys* adults to low temperatures has been previously linked to increased fecundity [42], while *Podisus* spp. displayed a higher production of offspring after colder winters as well [27].

## 5. Conclusions

In conclusion, our findings confirm the incidence of a facultative reproductive diapause of females of *H. halys* in N. Greece and provide valuable information on the phenology and reproductive physiology of this pest in this region. Despite its widespread presence in our country [6,7,24,43], little research has been conducted to investigate the biological parameters of this insect. Our results suggest that overwintering females of *H. halys* do not initiate egg laying right away once they leave their overwintering sites, but rather need some additional time for diapause termination in order to mature reproductively. This information could be very important for the design and implementation of a successful IPM plan to manage this pest, as a “time-to-act” indication. Moreover, our findings comply with previous studies that showed increased fecundity of *H. halys* and other stink bugs after exposure to low temperatures [27,41], and mark our study as a step forward towards assessing the risk of feeding damage of this pest to early season crops, based on the previous winter weather conditions.

## Figures and Tables

**Figure 1 insects-13-00866-f001:**
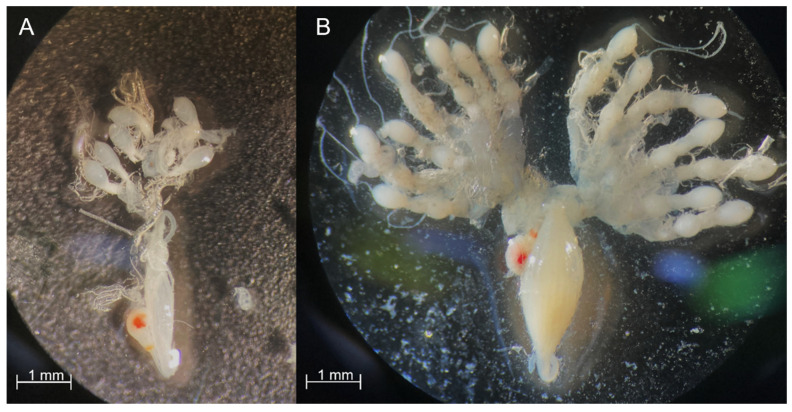
*Halyomorpha halys* female reproductive system. (**A**) Rank 2: previtellogenic female, more than one immature oocytes per ovariole, spermatheca clear and skinny. (**B**) Rank 3: vitellogenic female (control), at least one mature oocyte per ovariole, spermatheca distended (Images 2.5× magnification).

**Figure 2 insects-13-00866-f002:**
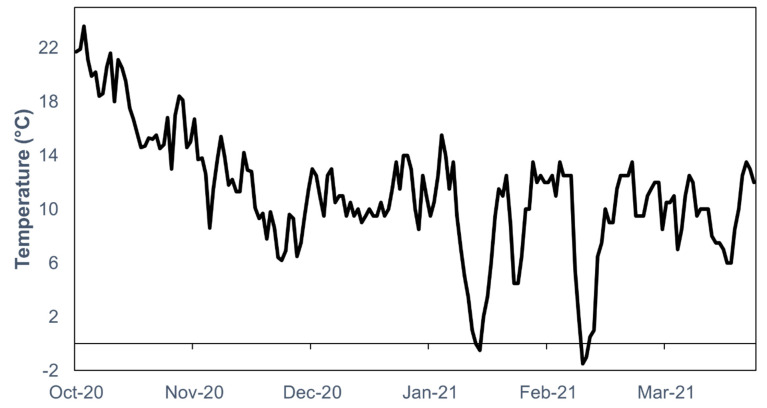
Average daily temperature (°C) in Thermi, Thessaloniki, Greece during the months of October, November, December 2020, and January, February and March 2021. Data recorded with HOBO U23 Pro v2 Temperature/Relative Humidity Data Logger (Onset Computing, Bourne, MA, USA).

**Figure 3 insects-13-00866-f003:**
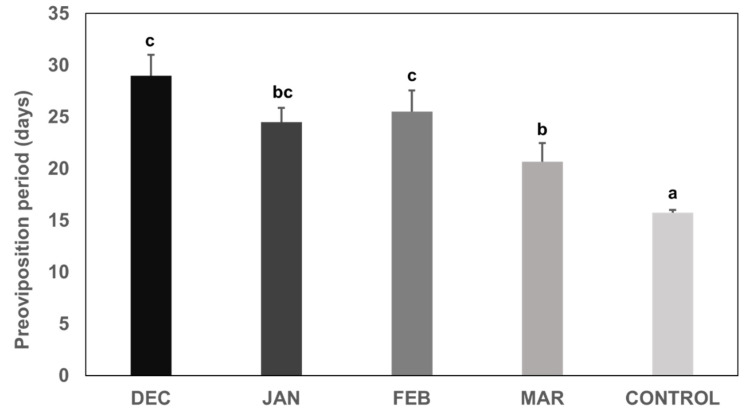
Preoviposition period (mean ± S.E., *n* = 60) of Halyomorpha halys females that overwintered in natural outdoor conditions until December 2020, January, February and March 2021 and were then transferred to 26 °C. A laboratory colony maintained at 26 °C was used as control. Different letters above bars indicate statistically significant differences (*p* < 0.001).

**Figure 4 insects-13-00866-f004:**
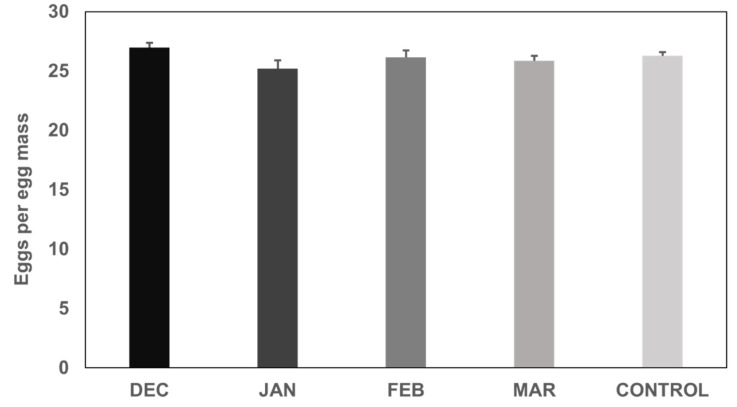
Number of eggs per egg mass (mean ± S.E., *n* = 60) laid by *Halyomorpha halys* females that overwintered in natural outdoor conditions until December 2020, January, February and March 2021 and were then transferred to 26 °C. A laboratory colony maintained at 26 °C was used as control. No statistically significant differences were observed (*p* = 0.165).

**Figure 5 insects-13-00866-f005:**
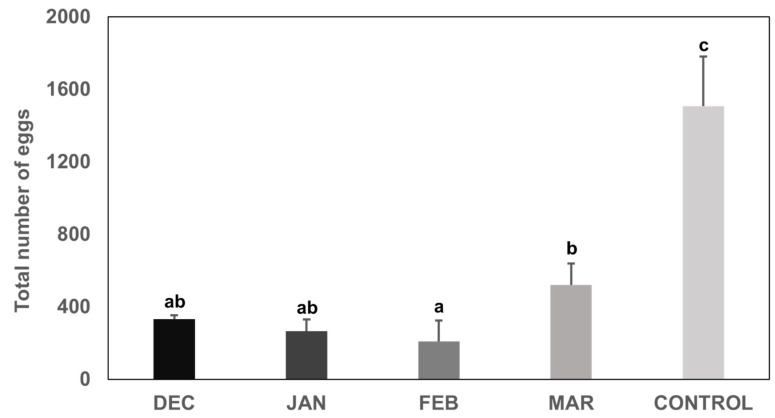
The total number of eggs (mean ± S.E. of 6 replicates) laid by *Halyomorpha halys* females (*n* = 10) that overwintered in natural outdoor conditions until December 2020, January, February and March 2021 and were then transferred to 26 °C. A laboratory colony maintained at 26 °C was used as control. Different letters above bars indicate statistically significant differences (*p* < 0.001).

## Data Availability

All the data is contained within the article.

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
