# Peer review of "Females of *Halyomorpha halys* (Hemiptera: Pentatomidae) Experience a Facultative Reproductive Diapause in Northern Greece"

_insects, 2022, doi:10.3390/insects13100866_

Round 1
Reviewer 1 Report
This paper presents a study of latency to oviposition and egg number of overwintering stink bugs. The experiment is very straightforward and the results are very clear. Females removed from overwintering conditions earlier in the winter demonstrate greater reproductive diapause and tend to lay fewer eggs. The paper is well written. I have a few minor questions and comments:
Line 118. I assume the control colony was maintained at optimum conditions but you should be stated explicitly.
Line 165. When were trials with the control colony performed? Did they correspond to the Dec-March dates of your other treatment groups?
Author Response
Line 118. I assume the control colony was maintained at optimum conditions but you should be stated explicitly.
We would like to thank the reviewer for their comments and additions. The control colony was indeed maintained at optimum (standard) laboratory conditions and this detail has been added in-text.
Line 165. When were trials with the control colony performed? Did they correspond to the Dec-March dates of your other treatment groups?
Yes, the control treatments were conducted in the same months as the other treatment groups. Individuals of the laboratory colony were selected monthly and maintained in groups of 10 males and 10 females in mesh cages (30x30x30 cm), just as in the other treatments.
Reviewer 2 Report
Diapause is an interesting phenomenon in insects and it is important for the insect growth and development. In this MS, the authors investigated the diapause of the female Halyomorpha halys in Northern Greece by determining the preoviposition period in different groups. This result will provide some information for the understanding the insect diapause. However, some issues should be concerned.
1. In the abstract and discussionm,the novelty of this experiment shoud be focused on.
2. Line 148, —- the following weather 147 conditions:18.1 ± 0.5°C (mean ±SEM) during the warmest month (October) and 8.2 ± 0.8°C 148 (mean ±SEM) during the coldest month (January), how about the photoperiod?
3. How the authors confirmed the developmental stages of females were same In Bioassay section (line 110)?
4. It Is reported that the diapause is also closed with the photoperiod in some insects, the effect of photoperiod on the diapause of Halyomorpha halys has been studied? Compared with the temperature, which one Is more important?
5. If some additional experiments were performed to explain the molecular mechanism, it will be best.
Author Response
- In the abstract and discussion, the novelty of this experiment shoud be focused on.
We would like to thank the reviewer for this comment. The novelty of our experiment was furtherly addressed in the abstract and discussion. The conclusions also focus on the novelty of our findings.
- Line 148, —- the following weather 147 conditions:18.1 ± 0.5°C (mean ±SEM) during the warmest month (October) and 8.2 ± 0.8°C 148 (mean ±SEM) during the coldest month (January), how about the photoperiod?
We would like to thank the reviewer for this valuable comment. Indeed, photoperiod is an important factor of diapause in insects. However, we are not referring to photoperiod in our manuscript, as it was not used as a factor in our experiments. Our experiments were not designed to study the effect of different photoperiods on diapause of this insect.
- How the authors confirmed the developmental stages of females were same In Bioassay section (line 110)?
We would like to thank the reviewer for this question. As stated by Nielsen et al. (2008), females of H. halys enter diapause reproductively immature. The fact that we collected individuals from the overwintering generation, as well as the finding that they were reproductively immature (dissections of females were conducted in 10 replicates to characterize the reproductive stage, as stated in-text: lines 127-133), makes us confident that all individuals used in our experiments were in the same developmental stage.
- It Is reported that the diapause is also closed with the photoperiod in some insects, the effect of photoperiod on the diapause of Halyomorpha halys has been studied? Compared with the temperature, which one Is more important?
Photoperiod is indeed an important factor of diapause in insects. The effect of photoperiod on diapause of H. halyshas been previously studied in several locations (Nielsen et al., 2016;2017; Cira et al., 2018). However, the importance of these two factors cannot be compared, as they induce different biological changes in insects. Furthermore, our study was not designed to investigate the effect of photoperiod on diapause of H. halys, and that is why we are focusing on the temperature profile of Northern Greece.
- If some additional experiments were performed to explain the molecular mechanism, it will be best.
We would like to thank the reviewer for this interesting suggestion. We are going to take this into consideration for further experiments.
Reviewer 3 Report
Dear Authors,
Find my comments in the attached and below.
Females of Halyomorpha halys (Hemiptera: Pentatomidae) experience a facultative reproductive diapause in Northern Greece
Comments to Authors
Major comments: The study contains good information but the amount of data is not enough and there are some flaws and inconsistencies in the experimental approach.
a) The data come from one year. For such type of studies on diapause, data from more than 1 year are mainly preferred to support any statement. Environmental conditions may differ dramatically from year to year, especially if we consider climate change effects (heatwaves, etc.).
b) Measuring preoviposition period is a good indicative proxy of reproductive diapause, but not enough to safely prove it. Conclusions should be supported by dissection of females and investigation of mature oocytes in each treatment.
c) Why females were hosted with males? Doesn’t male presence affect their fecundity?
c) Data from the egg hatch are MISSING from the results. N= number of individuals for each treatment is also MISSING. That type of information is fundamental.
d) Information on the control colony is limited. There is literally nothing known for the control.
e) Discussion is premature, repetitive of general information of previous studies and in several cases inaccurate.
E.g. Line 242 ‘temperature is known as the main abiotic factor to directly affect fecundity and oviposition in insects {Kim D.S. and Lee J.H. Oviposition model of Carposina sasakii (Lepidoptera: Carposinidae). Ecological Modelling 2003, 162, 145– 363 153.}. What is the relevance of that reference for your study and how can you support such general strong statements by using a single reference.
Carposina sasakii overwinters as larvae in the soil and has nothing to do with your study, so it should be used in first instance.
f) What new does this study provide to the scientific community that was not known before?
Specific comments: See specific comments in the text.
Line 11: oviposition rates
Lines 97-108: a) Overwintering should be justified to eliminate any doubts? How can you be sure that those individuals were overwintering? Simply because collected at overwintering sites? Either a REF is required or give more details on the environmental conditions (photoperiod & temp) in that region and in that specific time period, which would justify the term overwintering
b) Why those overwintering individuals haven’t been held in their initial site (i.e. in Naoussa area) and transferred to Thessaloniki? And why haven’t been collected initially from Thessaloniki? Is it because of logistics or is there a specific reason for this? Google maps shows that those 2 areas are distant.
c) When in October? Specify (beginning, mid or end of October). All together? Within a week or across the whole month?
Lines 110-124: a) What is the reason of starting your experiment on 30 of December?
Why didn’t it start from 1st of January?
It is not specified whether the exit date was the same for all the 6 replicates of each treatment or not. Was there a pattern/frequency in the exit date? Did it happen every stable number of days or all of them were transferred at the end of the calendar month?
If transfer to chamber room was taking place in pre-defined intervals (e.g. 5 days) then there is not consistency for month December and the rest of the months.
b) A reference justifying that these are the “optimum” conditions should be included. Definitely 26°C, 111 60% RH, 16L:8D h photoperiod are good conditions for insect development but optimum needs more details and justification. If there is not a reference, then omit the word optimum and use stable conditions within the optimum range.
c) More details for “established laboratory colony” are required. Following which protocol? Fed with what? generation in lab? Collected from where; And many more details are required. The reader needs to understand what were the exact conditions in the control so as to compare with the rest of treatments
Lines 126-132: 10 individuals out of 3000 collected is a pretty tiny sample. The number is not so convincing. More adults could and should have been dissected.
Did you examine the spermatheca as well to assure virginity in females?
Is there a specific reason of why legs and wings were removed and why this needs to be mentioned? I don’t see that this information adds anything.
Lines 241-255: …..”temperature is known as the main abiotic factor to directly affect fecundity and oviposition in insects”
Not always… Very absolute and vague statement. For some species optimum temperature conditions are not enough (e.g. protein requirement).
Actually, that statement is contradicted with what is been referred a few sentences below. This paragraph needs to be constructed in a more coherent way

Author Response
Dear reviewer,
thank you for providing your insight on our manuscript. We have revised our manuscript according to your comments:
Comments to Authors
Major comments: The study contains good information but the amount of data is not enough and there are some flaws and inconsistencies in the experimental approach.
- a) The data come from one year. For such type of studies on diapause, data from more than 1 year are mainly preferred to support any statement. Environmental conditions may differ dramatically from year to year, especially if we consider climate change effects (heatwaves, etc.).
-While collecting H. halys individuals to establish and refresh our laboratory colony (over the past 4 years- since 2018), we have made field observations based on the dispersion and deviation of the different developmental stages of this insect in the field. We observed that in Central Macedonia (where our collections took place) in the start of October, adults seek for sheltered locations to overwinter from which they emerge and become active in April, which has been also confirmed in other locations (Nielsen et al. (2008; Inkley, 2012; Lee et al., 2013;2014; Cira et al. 2016;2018; Bergh et al., 2017). Every year we collected adults (in the start of October) and transferred them to our laboratory in Thermi, Thessaloniki to refresh our established colony. We observed that adult females that were newly transferred to stable laboratory conditions (26°C, 60% RH, 16L:8D h photoperiod) following their exposure to natural outdoor conditions of Northern Greece, needed more time to lay their first egg mass. Our hypothesis led us to design the experiment that is presented in our study and we believe our results are solid enough to support our claims. Regarding the very interesting comment of the reviewer about climate change, we should take into account that we are investigating an invasive species, newly established in Greece and any changes in its behavior or biology because of climate change would be evident in the long term. Nonetheless, the investigation of climate change effects on the biology of insects is a different type of study that needs a different approach, which was not our purpose currently.
- b) Measuring preoviposition period is a good indicative proxy of reproductive diapause, but not enough to safely prove it. Conclusions should be supported by dissection of females and investigation of mature oocytes in each treatment.
-We would like to thank the reviewer for this statement. Females were dissected prior each treatment to ensure they were reproductively immature. After they were used for the experiment, we preferred to use optical observations of mating in each replicate as an indication of sexual maturity, which was furtherly evidenced by the oviposition of the female individuals.
- c) Why females were hosted with males? Doesn’t male presence affect their fecundity?
-It is not clear if the reviewer is asking about the individuals that were kept in natural outdoor conditions, or the ones in the treatments. If the question is about the overwintering individuals that were kept outside prior to the experiment, females were hosted with males because we wanted to simulate the overwintering conditions that occur naturally. If the question is about the treatments, females were hosted with males because we wanted to investigate the preoviposition period and fecundity of female individuals. To our knowledge, there hasn’t been any data on H. halys reproducing asexually, so naturally we thought that male presence is needed for reproduction.
- c) Data from the egg hatch are MISSING from the results. N= number of individuals for each treatment is also MISSING. That type of information is fundamental.
-Indeed, that type of information is fundamental. The number of individuals for each treatment is mentioned in-text: Lines 112-114 - 10 males and 10 females were randomly selected (…) A total of 6 replicates were conducted for each month. The total number of individuals were 120. This detail has been added to the text and Figure 3.
-Data from egg hatching has been added as supplementary material. We did not observe any significant differences in the hatchability of the eggs among each treatment and that is why we did not include this data in the manuscript.
- d) Information on the control colony is limited. There is literally nothing known for the control.
-We would like to thank the reviewer for this addition. We included details on the laboratory colony rearing methods.
- e) Discussion is premature, repetitive of general information of previous studies and in several cases inaccurate.
E.g. Line 242 ‘temperature is known as the main abiotic factor to directly affect fecundity and oviposition in insects {Kim D.S. and Lee J.H. Oviposition model of Carposina sasakii (Lepidoptera: Carposinidae). Ecological Modelling 2003, 162, 145– 363 153.}. What is the relevance of that reference for your study and how can you support such general strong statements by using a single reference.
Carposina sasakii overwinters as larvae in the soil and has nothing to do with your study, so it should be used in first instance.
-The above study was referenced by Baek et al. (2017) in their study “Temperature-dependent development and oviposition models of Halyomorpha halys (Hemiptera: Pentatomidae)”. However, since the reviewer poses an interesting dispute about the relevance of the referenced study, we replaced it with two more relevant ones (about hemipterans) that express the important effect of temperature on several biological characteristics in insects, including oviposition and fecundity.
- f) What new does this study provide to the scientific community that was not known before?
-Halyomorpha halys is a pest that has widely spread in Greece since 2011 (Milonas and Partsinevelos, 2014; Andreadis et al., 2018; 2022; Corsini-Foka et al., 2021). However, its biology has not been adequately studied. Our study provides information on the biology and reproductive physiology of this serious polyphagous pest in the region of N. Greece, where many of its primary hosts are cultivated. Knowledge on this species’ oviposition behavior and fecundity is crucial for the implementation of management strategies.
Specific comments: See specific comments in the text.
Line 11: oviposition rates
-The word rates has been added in Line 11.
Lines 97-108: a) Overwintering should be justified to eliminate any doubts? How can you be sure that those individuals were overwintering? Simply because collected at overwintering sites? Either a REF is required or give more details on the environmental conditions (photoperiod & temp) in that region and in that specific time period, which would justify the term overwintering
-As stated by Nielsen et al. (2008), adults of H. halys overwinter in a non-feeding, non-reproductive state. In the mid- Atlantic US states, adults overwinter in concealed, sheltered locations, underneath bark of trees or human-made structures (Inkley, 2012), within which thousands of individuals cluster together (Bergh et al., 2017). Bergh et al. (2017) also adds: “As in its native Asian range (Lee et al., 2013), adult H. halys in the eastern USA disperse to protected overwintering sites in late September and October, within which they form aggregations of a few to many thousands of individuals. These include natural and anthropogenic harborages, such as under loose bark of fallen trees (Lee et al., 2014) or rock piles and rocky outcroppings, human-made structures, and many other sites associated with human activity.” Moreover, in a study by Cira et al. (2018), it is mentioned: “Regardless of location, aggregating in protected sites to overwinter allows H. halys to cope with thermally unfavorable periods that occur in temperate climates (Cira et al. 2016).” The above information is included in the introduction section of our manuscript.
-The fact that we found and collected only adults (no nymphs were found) on the 6th and 7th of October from protected locations (inside abandoned houses), where they had formed clusters, as well as the fact that they were dissected and evident to be reproductively immature, makes us confident that the individuals collected were in fact the overwintering generation.
- b) Why those overwintering individuals haven’t been held in their initial site (i.e. in Naoussa area) and transferred to Thessaloniki? And why haven’t been collected initially from Thessaloniki? Is it because of logistics or is there a specific reason for this? Google maps shows that those 2 areas are distant.
-Individuals were held in Thermi (Thessaloniki) because our lab is located there. It was simply a matter of logistics for the conducting of the experiment.
-Naousa was the location where we found a large population of H. halys adults, and that is why we collected the individuals from this area.
- c) When in October? Specify (beginning, mid or end of October). All together? Within a week or across the whole month?
-We would like to thank the reviewer for this comment. Individuals were collected within 2 days on October 6th and 7th, 2020. This detail has been added in-text.
Lines 110-124: a) What is the reason of starting your experiment on 30 of December?
Why didn’t it start from 1st of January?
-There is no specific reason for this. Nonetheless, we don’t believe 2 days would inflict any major changes on our experiment.
It is not specified whether the exit date was the same for all the 6 replicates of each treatment or not. Was there a pattern/frequency in the exit date? Did it happen every stable number of days or all of them were transferred at the end of the calendar month?
If transfer to chamber room was taking place in pre-defined intervals (e.g. 5 days) then there is not consistency for month December and the rest of the months.
-The 6 replicates were conducted monthly in 30-day intervals (December 30, January 29, February 28, March 30).
- b) A reference justifying that these are the “optimum” conditions should be included. Definitely 26°C, 111 60% RH, 16L:8D h photoperiod are good conditions for insect development but optimum needs more details and justification. If there is not a reference, then omit the word optimum and use stable conditions within the optimum range.
-We agree with the reviewer on this statement and per their suggestion, we replaced the word “optimum” with “stable”.
- c) More details for “established laboratory colony” are required. Following which protocol? Fed with what? generation in lab? Collected from where; And many more details are required. The reader needs to understand what were the exact conditions in the control so as to compare with the rest of treatments
-This addition is indeed very important and we addressed it in-text (please see above comment about the control treatment as well).
Lines 126-132: 10 individuals out of 3000 collected is a pretty tiny sample. The number is not so convincing. More adults could and should have been dissected.
Did you examine the spermatheca as well to assure virginity in females?
-This is an omission in the manuscript, and we would like to thank the reviewer for pointing that out. 10 individuals were dissected prior each treatment. This detail has been added to the text.
-Yes, we also examined the spermatheca to assure virginity in the collected females prior to using them in the experiment. The spermatheca can be seen clear and skinny in Figure 1 (A). A description of the spermatheca was added in Figure 1 legend.
Is there a specific reason of why legs and wings were removed and why this needs to be mentioned? I don’t see that this information adds anything.
-Wings and legs were removed prior dissection for ease of the process. We mention this to provide detailed information on the method of dissection.
Lines 241-255: …..”temperature is known as the main abiotic factor to directly affect fecundity and oviposition in insects”
Not always… Very absolute and vague statement. For some species optimum temperature conditions are not enough (e.g. protein requirement).
Actually, that statement is contradicted with what is been referred a few sentences below. This paragraph needs to be constructed in a more coherent way
-We removed this reference entirely and replaced it with two more relevant ones to our study (see above comment about the discussion part of our manuscript as well). This part of the discussion was rewritten per the reviewer’s request.
Round 2
Reviewer 2 Report
the molecular mechanism of insect diapause can be mentioned in introduction.
Author Response
Lines 81-88 were added to the introduction section referring to the molecular mechanism of insect diapause.
Reviewer 3 Report
Dear authors,
Most of the comments have been addressed in the revised ms. I still believe that there is room for further elaboration in the discussion section if you intend to attract readers' attention .
Author Response
Dear reviewer,
Thank you for taking the time to review our manuscript. We have revised it accordingly, after receiving the editors’ notes as well.